# Ab Initio Thermoelasticity of Liquid Iron-Nickel-Light Element Alloys

**Hiroki Ichikawa †** and **Taku Tsuchiya \*,†** 

Geodynamics Research Center, Ehime University, Matsuyama 790-8577, Japan; takutsuchiya@gmail.com
\* Correspondence: tsuchiya.taku.mg@ehime-u.ac.jp
† These authors contributed equally to this work.

**Abstract:** The earth's core is thought to be composed of Fe-Ni alloy including substantially large amounts of light elements. Although oxygen, silicon, carbon, nitrogen, sulfur, and hydrogen have been proposed as candidates for the light elements, little is known about the amount and the species so far, primarily because of the difficulties in measurements of liquid properties under the outer core pressure and temperature condition. Here, we carry out massive ab initio computations of liquid Fe-Ni-light element alloys with various compositions under the whole outer core $P$, $T$ condition in order to quantitatively evaluate their thermoelasticity. Calculated results indicate that Si and S have larger effects on the density of liquid iron than O and H, but the seismological reference values of the outer core can be reproduced simultaneously by any light elements except for C. In order to place further constraints on the outer core chemistry, other information, in particular melting phase relations of iron light elements alloys at the inner core-outer core boundary, are necessary. The optimized best-fit compositions demonstrate that the major element composition of the bulk earth is expected to be CI chondritic for the Si-rich core with the pyrolytic mantle or for the Si-poor core and the $(Mg,Fe)SiO_3$-dominant mantle. But the H-rich core likely causes a distinct Fe depletion for the bulk Earth composition.

**Keywords:** ab initio molecular dynamics; high-pressure thermoelasticity; outer core chemistry

## 1. Introduction

The earth's core is thought to be composed of Fe-Ni alloy including substantially large amounts of light elements. These light elements account for observed density deficits of ~10% for the liquid outer core and ~5% for the solid inner core [1–7]. Determination of the light element (LE) composition of the outer core (OC) has long been one of the central research topics in the deep earth sciences. The density ($\rho$) and adiabatic bulk ($K_T$) and shear ($K_S$) moduli of iron and iron-LE alloys are key to interpreting seismological observations and then constructing a compositional model of the core [5,8]. However, those of the liquid states at the OC pressure ($P$) and temperature ($T$) (from ~136 to ~329 GPa and from ~4000 to ~6000 K) are still limitedly clarified experimentally. So far, static experiments have been performed up to less than 100 GPa [9–11]. Higher-$P$ behavior of liquid iron was on the other hand investigated by shock wave experiments in multi-Mbar condition [2,12–16]. The temperature, however, changes along the principal Hugoniot and dramatically increases with increasing pressure to more than 8000 K at the pressure of the inner core (IC)-OC boundary ($P_{ICB}$) of ~329 GPa, which is far higher than the expected actual ICB temperature ($T_{ICB}$) of ~5000–6000 K [17–22]. Experimental determination of thermoelasticity of liquid iron alloys in the whole $P$, $T$ condition of the earth's OC thus remains technically impractical.

In contrast, ab initio molecular dynamics (AIMD) simulations have been widely applied to clarify $\rho$ and P-wave velocity ($V_P$) of liquid iron and iron-LE alloys at the OC conditions in order to constrain

the OC composition by interpreting seismological observations. These parameters for the Fe-O, Fe-Si, Fe-S, Fe-C, Fe-Ni, and Fe-Si-O system were calculated [23–25]. However, the data points in these studies were limited; two particular compositions of $Fe_{0.82}Si_{0.10}O_{0.08}$ and $Fe_{0.79}Si_{0.08}O_{0.13}$ only were considered [24] and two particular pressures of the core-mantle boundary (CMB) and ICB only were considered [23]. In particular, in the latter, empirical pressure corrections of 10 GPa and 8 GPa were adopted at the CMB and ICB respectively, though the optimized OC compositions are essentially sensitive to these corrections. Meanwhile, some studies have been performed throughout the whole OC $P$, $T$ conditions for pure Fe [7,26], Fe-S [27], and Fe-H [28]. However, different formulations were employed to model their thermal equations of state, making a quantitative comparison of the reported thermoelasticity not easy.

In this study, ab initio MD simulations are performed for binary and ternary Fe-Ni-LE alloys with several different LE and Ni fractions from ~100 to ~450 GPa and from 4000 to 8000 K. Equations of state (EoS) and thermoelasticity are then analyzed for each alloy through the same internally consistent way [7]. Using modeled thermoelasticity, we optimize light element compositions for each alloy as a function of the $T_{ICB}$ and discuss the possible OC composition.

## 2. Results and Discussion

### 2.1. Effects of LE on the Thermoelasticity of Liquid Iron

Calculations with several different LE concentrations clarify systematic trends on the effects of LEs on the thermoelasticity of liquid Fe. Incorporations of LEs always decrease $\rho$ but increase $V_P$ (Table 1), but trends are different depending on the type of LE. It is found that incorporations of larger Si and S atoms have only marginal effects on the volume (volume per atom), then the EoS is nearly unchanged (Figure S1). In contrast, incorporations of smaller O, C, and in particular H atoms reduce the volume considerably in the whole OC $P$ range. These are related to the fact that the Fe-Si and S alloys are so-called substitutional-type, while the Fe-O, C, and H alloys are interstitial-type as recognized generally in lower $P$ condition.

**Table 1.** Effects of light element (LE) incorporation on $V_P$ and $\rho$ of liquid Fe calculated at the $P_{CMB}$ and 4000 K and at the $P_{ICB}$ and 5300 K. $X_{LE}$ represents the fraction of LE in atom%.

| *P, T* Condition | | **O** | **Si** | **S** | **C** | **H** |
|---|---|---|---|---|---|---|
| $P = P_{CMB}$ $T = 4000$ K | $\frac{\partial \ln V_P}{\partial X_{LE}}$ | 0.05(1) | 0.13(1) | 0.06(1) | 0.16(1) | 0.02(1) |
| | $\frac{\partial \ln \rho}{\partial X_{LE}}$ | −0.34(1) | −0.51(1) | −0.41(1) | −0.30(1) | −0.24(1) |
| | $\frac{\partial \ln V_P}{\partial \ln \rho}$ | −0.14(1) | −0.26(1) | −0.16(1) | −0.54(1) | −0.10(1) |
| $P = P_{ICB}$ $T = 5300$ K | $\frac{\partial \ln V_P}{\partial X_{LE}}$ | 0.09(1) | 0.21(1) | 0.16(1) | 0.20(1) | 0.07(1) |
| | $\frac{\partial \ln \rho}{\partial X_{LE}}$ | −0.31(1) | −0.48(1) | −0.38(1) | −0.31(1) | −0.21(1) |
| | $\frac{\partial \ln V_P}{\partial \ln \rho}$ | −0.29(1) | −0.44(1) | −0.42(1) | −0.63(1) | −0.35(1) |

Because of these volume reductions, $\rho$ variations associated with the O and H incorporations are smaller than those expected from the small masses. As a result, the effects of Si and S incorporations on $\rho$ are larger than those of O and H incorporations (Table 1). These behaviors are consistent with a recent study reporting structural and dynamical properties of Fe-LE alloy liquids [29]. A similar tendency is seen in $V_P$, but the systematics is less pronounced since the effects of LEs on $\rho$ and $K_S$ are partially cancelled. Perturbation ratios ($\partial \ln V_P / \partial \ln \rho$) are sometimes referred to discuss the chemical heterogeneity in Earth's deep interior [30,31]. In the present cases, absolute values of this ratio are always smaller than 1, indicating that the effects of LE incorporations are always much larger in $\rho$ than in $V_P$.

## 2.2. Optimized Compositions

Misfits in $\rho$ and $V_P$ between the Fe-Ni-$X$ liquid alloys and the preliminary reference earth model (PREM) [32] are then evaluated as $\sum\left[\left(\frac{\rho-\rho_{PREM}}{\rho_{PREM}}\right)^2 + \left(\frac{V_P-V_{P_{PREM}}}{V_{P_{PREM}}}\right)^2\right]$ (Figure 1), where the summation is taken over the whole OC pressure range. It is clearly demonstrated that the misfits are sensitive to the LE concentration and temperature but not so sensitive to the Ni concentration. The best-fit compositions along two adiabats ($T_{ICB}$ = 5000 K and 6500 K) with three different Ni/(Fe + Ni) ratios, which can be defined by the minima of misfits, are listed in Table 2 with misfits and the $\rho$ and $V_P$ of best-fit compositions along two adiabats are shown in Figure 2.

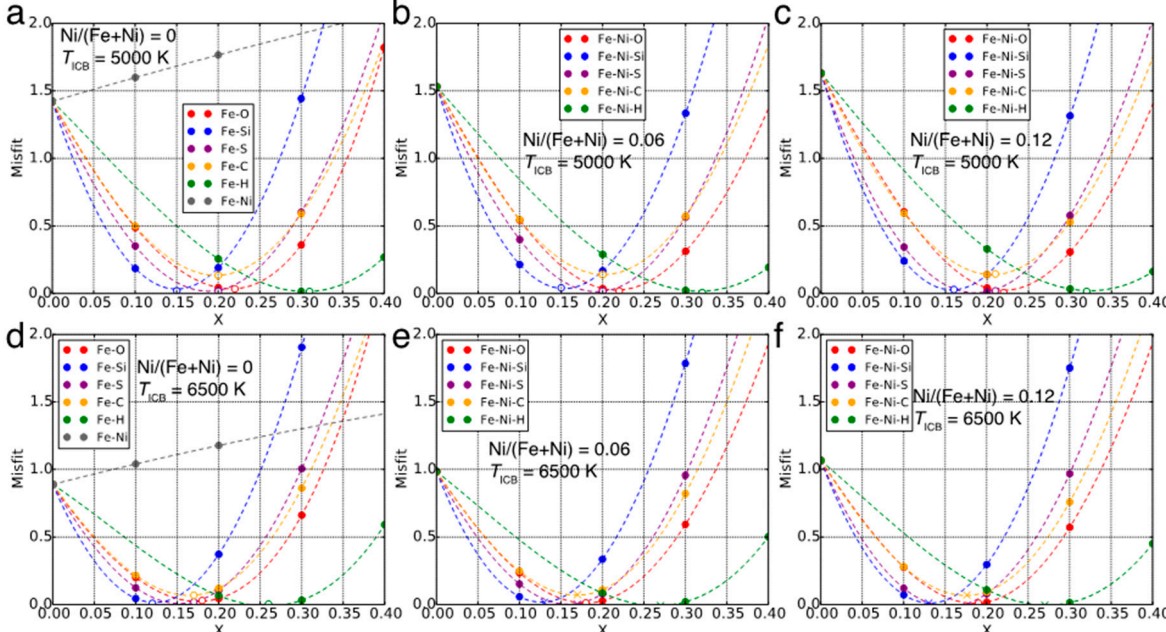

**Figure 1.** The misfit from the value of the preliminary reference earth model (PREM) as a function of atomic fraction of LEs ($X_{LE}$). Three different Ni fractions, 0 (**a**,**d**), 0.06 (**b**,**e**), and 0.12 (**c**,**f**) and two different $T_{ICB}$ of 5000 K (**a**–**c**) and 6500 K (**d**–**f**) are examined. Filled circles are the results of molecular dynamics (MD) and dashed lines are the cubic spline interpolations. The atomic fractions at the minima correspond to the best-fit LE concentrations. Open circles indicate the misfits obtained from MD with the best-fit compositions, which are in good agreement with the minima of spline interpolations. Errors in the misfits originated in the fitting procedures are comparable to the size of symbols.

Figures 1 and 2 and Table 2 indicate that among the best-fit compositions, the misfit of the Fe(-Ni)-C system is distinctly large. This suggests that carbon could be eliminated from the major LE in the OC, though there is possibility that the situation could change in ternary or higher-order multicomponent systems. In contrast, the misfits of the best-fit compositions of the other LEs have marginal differences, which are almost indistinguishable from each other within the computational uncertainty (shaded regions in Figure 2). There have been some previous studies which constrained the OC composition within the similar manner, taking two LEs into account, suggesting an oxygen-depleted OC [12] or oxygen-rich OC (3.7 wt. % O, 1.9 wt. % Si) [23]. However, according to the results of the present study, the difference between the misfits of best-fit composition models (Figure 1) is very small except for C, indicating that the information of $\rho$ and $V_P$ are insufficient to determine the OC composition uniquely. Therefore, some other information, e.g., melting phase relations, partitioning behavior between solids and liquids, the bulk earth (BE) compositional property and so on, are quite helpful to place further constraints on the LE composition, but all of these are not well understood at the moment.

**Table 2.** Best-fit compositions of binary and ternary alloys at $T_{ICB}$ = 5000 K and 6500 K. Misfit, Mg/Si, and Mg/Fe represent a misfit in $V_P$ and $\rho$ from the PREM, Mg/Si, and Mg/Fe ratios expected to the bulk earth with the pyrolytic mantle, respectively. Errors from the fitting procedures are represented in parentheses.

| $T_{ICB}$ | Best-Fit Composition | Misfit($\times 10^{-2}$) | Mg/Si | Mg/Fe |
|---|---|---|---|---|
| 5000 K | $Fe_{0.78}O_{0.22}$ | 1.8(1) | 1.25(1) | 1.03(1) |
| | $Fe_{0.85}Si_{0.15}$ | 2.7(1) | 1.06(1) | 1.04(1) |
| | $Fe_{0.81}S_{0.19}$ | 1.6(1) | 1.25(1) | 1.08(1) |
| | $Fe_{0.80}C_{0.20}$ | 11.2(1) | 1.25(1) | 1.01(1) |
| | $Fe_{0.70}H_{0.30}$ | 1.9(1) | 1.25(1) | 0.97(1) |
| | $Fe_{0.73}Ni_{0.05}O_{0.22}$ | 0.8(1) | 1.25(1) | 1.10(1) |
| | $Fe_{0.80}Ni_{0.05}Si_{0.15}$ | 1.3(1) | 1.06(1) | 1.10(1) |
| | $Fe_{0.76}Ni_{0.05}S_{0.19}$ | 0.6(1) | 1.25(1) | 1.14(1) |
| | $Fe_{0.75}Ni_{0.05}C_{0.20}$ | 11.8(1) | 1.25(1) | 1.07(1) |
| | $Fe_{0.64}Ni_{0.04}H_{0.32}$ | 1.1(1) | 1.25(1) | 1.03(1) |
| | $Fe_{0.69}Ni_{0.09}O_{0.22}$ | 0.7(1) | 1.25(1) | 1.15(1) |
| | $Fe_{0.74}Ni_{0.10}Si_{0.16}$ | 1.4(1) | 1.05(1) | 1.17(1) |
| | $Fe_{0.71}Ni_{0.10}S_{0.19}$ | 1.2(1) | 1.25(1) | 1.21(1) |
| | $Fe_{0.7}Ni_{0.09}C_{0.21}$ | 10.4(1) | 1.25(1) | 1.13(1) |
| | $Fe_{0.6}Ni_{0.08}H_{0.32}$ | 0.9(1) | 1.25(1) | 1.09(1) |
| 6500 K | $Fe_{0.82}O_{0.18}$ | 4.8(1) | 1.25(1) | 1.02(1) |
| | $Fe_{0.88}Si_{0.12}$ | 1.7(1) | 1.09(1) | 1.02(1) |
| | $Fe_{0.85}S_{0.15}$ | 0.8(1) | 1.25(1) | 1.05(1) |
| | $Fe_{0.84}C_{0.16}$ | 7.7(1) | 1.25(1) | 1.00(1) |
| | $Fe_{0.74}H_{0.26}$ | 0.1(1) | 1.25(1) | 0.97(1) |
| | $Fe_{0.77}Ni_{0.05}O_{0.18}$ | 0.9(1) | 1.25(1) | 1.08(1) |
| | $Fe_{0.82}Ni_{0.05}Si_{0.13}$ | 1.8(1) | 1.08(1) | 1.08(1) |
| | $Fe_{0.79}Ni_{0.05}S_{0.16}$ | 0.2(1) | 1.25(1) | 1.12(1) |
| | $Fe_{0.78}Ni_{0.05}C_{0.17}$ | 7.5(1) | 1.25(1) | 1.06(1) |
| | $Fe_{0.69}Ni_{0.04}H_{0.27}$ | 2.3(1) | 1.25(1) | 1.02(1) |
| | $Fe_{0.71}Ni_{0.1}O_{0.19}$ | 2.2(1) | 1.25(1) | 1.15(1) |
| | $Fe_{0.77}Ni_{0.1}Si_{0.13}$ | 1.3(1) | 1.08(1) | 1.15(1) |
| | $Fe_{0.74}Ni_{0.1}S_{0.16}$ | 0.6(1) | 1.25(1) | 1.18(1) |
| | $Fe_{0.72}Ni_{0.1}C_{0.18}$ | 7.7(1) | 1.25(1) | 1.13(1) |
| | $Fe_{0.64}Ni_{0.09}H_{0.27}$ | 1.0(1) | 1.25(1) | 1.09(1) |

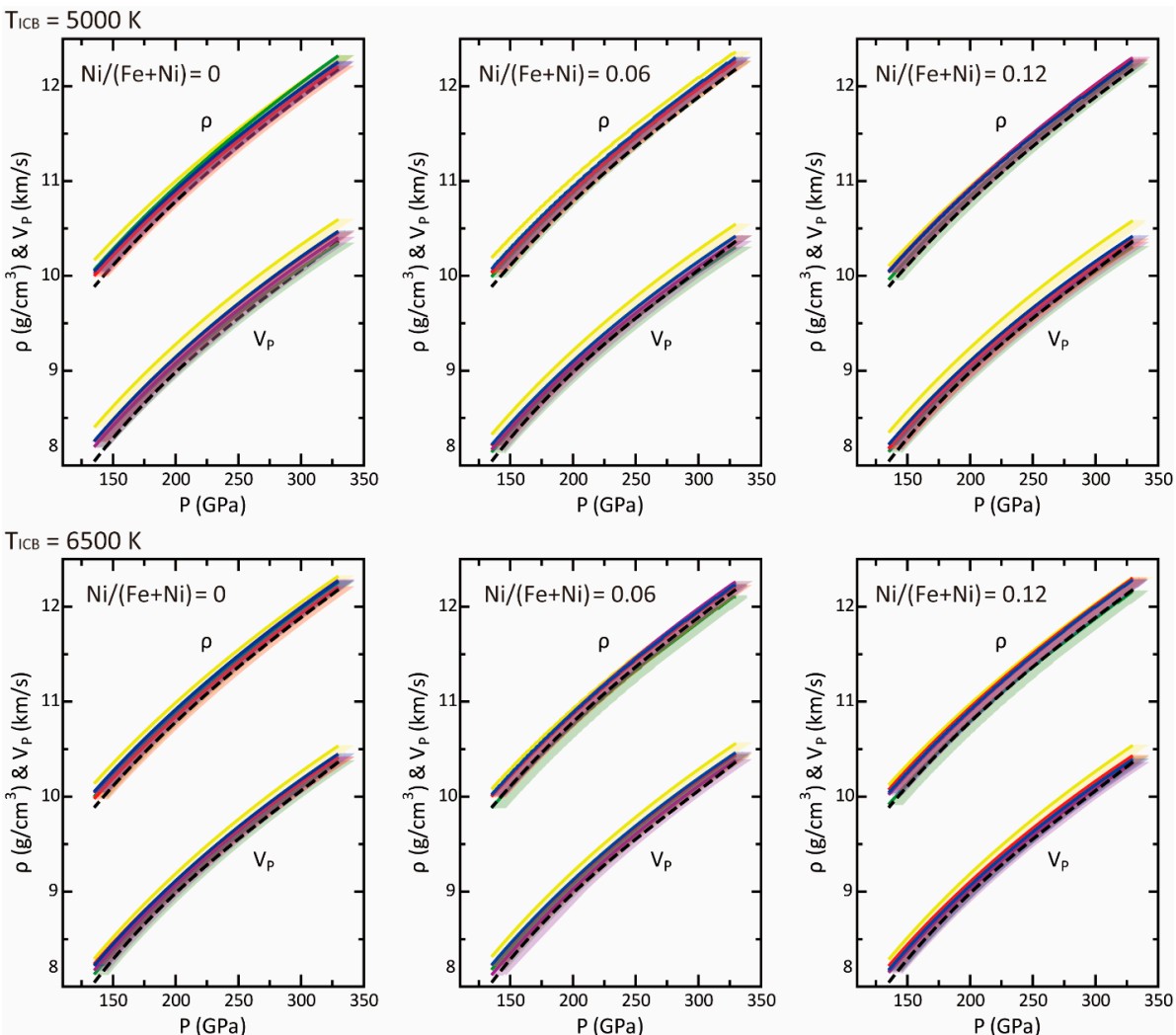

**Figure 2.** $\rho$ and $V_P$ for the best-fit compositions along two adiabats with $T_{ICB}$ = 5000 K and 6500 K (red, Fe-Ni-O; blue, Fe-Ni-Si; purple, Fe-Ni-S; yellow, Fe-Ni-C; green, Fe-Ni-H). Solid lines correspond to the results determined from raw thermoelasticity data and shaded regions correspond to the uncertainties in pressure with +10 GPa [33]. Dashed lines indicate the PREM values [32]. The $\rho$ and $V_P$ of almost all alloys overlap, indicating that these data only are insufficient to determine the OC composition uniquely.

The best-fit compositions vary depending on the setting of $T_{ICB}$ since the amount of LEs required to reproduce the PREM decreases for higher $T$ (Table 2). However, the misfit values are insensitive to temperature without any systematic variations, meaning that it is difficult to constrain $T_{ICB}$ through this optimization. Based on the calculated results at two different $T_{ICB}$, the best-fit compositions are represented as a function of $T_{ICB}$ within the first-order as follows:

$$X_O \text{ (atom\%)} = -2.60 \times 10^{-3} \, T_{ICB}(K) + 35.0 \text{ for O,} \tag{1}$$

$$X_{Si} \text{ (atom\%)} = -1.80 \times 10^{-3} \, T_{ICB}(K) + 24.4 \text{ for Si,} \tag{2}$$

$$X_S \text{ (atom\%)} = -2.27 \times 10^{-3} \, T_{ICB}(K) + 30.6 \text{ for S,} \tag{3}$$

and

$$X_H \text{ (atom\%)} = -3.33 \times 10^{-3} \, T_{ICB}(K) + 48.3 \text{ for H.} \tag{4}$$

These are found to change only marginally when Ni is incorporated. Here, the Fe(-Ni)-C systems are eliminated since they show larger misfits than others.

Since the $T_{ICB}$ should correspond to the freezing temperature of the OC liquid, melting phase relations of the Fe-LEs systems at the $P_{ICB}$ are quite important to place further constraints on the OC composition. There are however almost no available data with enough quality at the moment. Some experiments, though all were conducted at substantially lower pressures than $P_{ICB}$, suggest that the eutectic temperature of Fe-FeS system is more than 1000 K lower than the melting temperature of pure Fe [19,34], while solidus or eutectic temperatures are not quite different (within a few 100 K) in the Fe-FeSi [35] and Fe-O systems [36]. A large drop in the melting temperature might also be expected in the Fe-H system [37]. The $T_{CMB}$ is usually thought to be ~4000 K [38,39] and its adiabatic extrapolation leads to ~5200 K for the $T_{ICB}$ [7]. This might be ~1000 K lower than the $T_M$ of pure Fe at the $P_{ICB}$, suggesting S and H as the potential LEs in the OC. But even larger temperature drops could exist in the ternary or quaternary systems, so it is hard to exclude Si and O from the LE candidates based on this discussion.

Another point is the density jump across the ICB ($\Delta\rho_{ICB}$), which is reported seismologically to reach ~4.7% [32]. This observed $\Delta\rho_{ICB}$ is however too large to be reconciled simply by the solid–liquid transition of pure Fe[41]. Partitioning of LEs between solid and liquid phases is therefore thought to be required, namely LEs dissolving in the OC should strongly prefer liquid to solid. Again, nothing can be conclusive before the melting phase relations of the Fe-LEs systems are clarified at $P_{ICB}$, but extrapolations of experimental knowledge obtained at lower pressures suggest that the strong partitioning occurs in the Fe-O system [36] but not in the Fe-S [19,34], Fe-Si [35,40], and Fe-H systems [37].

In this study, we select the PREM model as a reference P-wave velocity and density of the OC. It is however well-known that the velocity structure of the earth's interior depends on the reference model. For example, the AK135 model [41,42] has the P wave velocity different from the PREM model in particular at the uppermost and lowermost outer core. Although contrasts between the AK135 and PREM reach ~0.11 km/s and ~0.065 km/s at the uppermost and lowermost part respectively, these make no significant changes in the insights obtained from our analyses.

## 2.3. Bulk Earth Composition

The BE composition can be affected by the LE compositions of the OC. We next examine what major element composition of the BE are lead from each best-fit composition for the OC (Table 2). In this modeling, the mantle composition is assumed to be the pyrolytic [43], and the IC and crust are ignored because of its negligibly small masses. The chemistry of the mantle, in particular of the lower mantle, is still under debate, but some recent studies similarly suggested the pyrolytic one might be reasonable [44,45].

Table 2 shows the Mg/Si and Mg/Fe ratios of the BE expected from the best-fit compositions for the core combined with the pyrolytic composition for the mantle, which are calculated using the total mass of the earth, weight and atomic % of major elements (Mg, Fe, Si, and O) in the pyrolytic model for the mantle and the optimized composition models for the core. CI chondrite, one of the major candidates of earth's building block, is known to have the Mg/Si and Mg/Fe ratios of ~1.05 and ~1.23, respectively [43]. Table 2 shows that this Mg/Si ratio is achieved only when Si is the major LE in the OC, but no case can explain the Mg/Fe ratio. Enstatite (EH) chondrite is another candidate of earth's building block, which is known to have the Mg/Si and Mg/Fe ratios of ~0.79 and ~0.91. Very Si-rich core and mantle are required to explain this small Mg/Si and Table 2 shows that such composition is incompatible with the observations of the OC. An Mg/Si value similar to CI chondrite, ~1.03, is proposed in the OCCAM model [46] with an Mg/Fe ratio of ~1.12. These ratios are close to the values expected for our Si-bearing best-fit compositions. In summary, Si is a geochemically plausible candidate for the major LEs in the core. But if the lower mantle is assumed to be MgSiO$_3$-dominant, the Mg/Si ratios of the BE expected with the best-fit composition for the core decreases by ~0.2. Then, the Mg/Si and Mg/Fe ratios of all the best-fit compositions except for Si and H-bearing cases match the

ratios of CI chondrite and OCCAM model. In this case, Si-rich and H-rich OC with MgSiO$_3$-dominant lower mantle lead to a too Si-rich and Fe-poor BE composition, respectively.

## 3. Conclusions

Ab initio thermoelasticity of Fe-Ni-LEs alloy liquids in the whole OC *P*, *T* condition indicates that all the LEs have the effects to decrease $\rho$ and increase $V_P$ of pure Fe, but the effects are counterintuitively larger for the Si and S incorporations than for the O, C, and H incorporations. Any best-fit alloy composition except the C-rich case can reproduce the $\rho$ and $V_P$ of the actual OC in the comparable level, so that the information of $\rho$ and $V_P$ only are insufficient to determine the OC composition uniquely. Melting phase relations and LE partitioning in the Fe-Ni-LE systems at the $P_{ICB}$ are therefore essential to place a tighter constraint on the OC chemistry. The Si-rich best-fit composition for the core with an assumption of the pyrolytic mantle predicts the CI chondritic BE composition, but the O and S-rich best-fit compositions for the core with the MgSiO$_3$-dominant mantle also lead to the similar chemistry for the BE. The H-rich best-fit composition however causes a distinct deficit of Fe for the BE. In future studies, it might be important to investigate correlations between LEs in higher-order multicomponent systems, which are ignored in this study.

## 4. Methods

### 4.1. Ab Initio Molecular Dynamics Simulations

To determine the *P-V-T* equation of state (EoS) of liquid iron-light element alloys, total internal energy (*E*) and total pressure (*P*) are calculated by means of the AIMD technique within the canonical (*NVT*) ensemble in the same manner as our previous study [7] using a PWSCF code [47] for electronic structure with an original implementation of the constant temperature molecular dynamics (MD) module [48]. The simulations are performed on binaries and ternaries, (Fe-Ni)$_{1-X}$O$_X$; (Fe-Ni)$_{1-X}$Si$_X$; (Fe-Ni)$_{1-X}$S$_X$; (Fe-Ni)$_{1-X}$C$_X$; (Fe-Ni)$_{1-X}$H$_X$, at different atomic fractions ($X_O \leq 0.5$, $X_{Si} \leq 0.3$, $X_S \leq 0.3$, $X_C \leq 0.3$, $X_H \leq 0.4$). Three Ni/(Fe + Ni) ratios of 0, 0.06 (consistent with the geochemically modeled value) [43], and 0.12 are examined.

The Newton's equation of motion is numerically integrated by using the velocity Verlet algorithm with time steps of 1 fs ($10^{-15}$ s) for the Fe-LE systems, which is the same for previous studies [7,17,33,49], and 0.5 fs for the Fe-H system. Some results (pure Fe and Fe$_{1-X}$O$_X$ systems) are, in part, already reported in the previous studies [7,49]. MD cells basically contain 50 atoms as in our previous study [7] but 100 atoms for the optimized compositions, and *T* is controlled by the kinetic energy scaling method. The validity of the cell size with 50–100 atoms for liquid iron can be seen in previous calculations [17,33], where a minor variation of the melting temperature of iron (~100 K) was found with changing the cell size from 67 to 980 atoms. Thermodynamic properties of liquid iron were also found to be sufficiently converged for this cell size.

For electronic structure calculations, we apply the generalized gradient approximation (GGA) [50] to the exchange correlation functional instead of the local density approximation (LDA). This is essential since many previous studies reported that GGA shows significant improvements over LDA when it comes to correctly describing ground-state properties and compression behaviors for iron [51,52]. The ultrasoft pseudopotential and plane-wave basis set are used to describe electronic structures. Here, an electronic configuration of $3s^23p^63d^{6.5}4s^14p^0$ is pseudized with a sufficiently small core radius of 2.0 a.u. for Fe; $2s^22p^4$, with a core radius of 1.5 a.u. for O; $3s^23p^4$, with a core radius of 1.7 a.u. for S; $3s^23p^23d^0$, with a core radius of 1.4 a.u. for Si; $2s^22p^2$, with a core radius of 1.1 a.u. for C; $1s^1$, with a core radius of 0.8 a.u. for H; and $3s^23p^63d^84s^24p^0$ with a core radius of 2.0 a.u. for Ni by the Vanderbilt scheme [53] with non-linear core corrections [54]. We apply a kinetic energy cutoff of 50 Ry and spin polarization is not taken into account. These conditions are already well tested in our previous calculations [7,49,55] and are fairly similar to those in calculations by other groups [24]. Liquids in principle have no periodic structure, thus the $\Gamma$ point only is sampled in our simulations. All the MD

simulations are conducted in *P,T* condition from ~80 to ~500 GPa and from 4000 to 8000 K (Figure S1), which covers the whole *P,T* range of the core. Standard deviations in calculated *T* and *P* are found to be typically ~50 K and ~3 GPa at 5000 K and ~130 GPa and ~120 K and ~6 GPa at 8000 K and ~400 GPa, respectively.

*4.2. EoS Analysis of Liquid Iron Alloys*

The calculated *E-P-V-T* relations of liquid iron alloys are analyzed using a single EoS model basically identical to the one in our previous study [7]. For the isothermal part at a reference temperature $T_0$, we used the Vinet (Morse-Rydberg) Equation,

$$P_{T_0}(V) = 3K_{T_0}\left(\frac{V}{V_0}\right)^{-\frac{2}{3}}\left[1 - \left(\frac{V}{V_0}\right)^{\frac{1}{3}}\right]\exp\left\{\frac{3}{2}\left(K'_{T_0} - 1\right)\left[1 - \left(\frac{V}{V_0}\right)^{\frac{1}{3}}\right]\right\} \tag{5}$$

Here, $K_{T_0}$ and $K'_{T_0}$ are the isothermal bulk modulus and its pressure derivative at zero pressure at $T_0$. The internal thermal energy is represented by a second-order polynomial of temperature with a volume dependent second-order coefficient,

$$E_{th}(V,T) = 3nk_B\left[T + e_0\left(\frac{V}{V_0}\right)^g T^2\right] \tag{6}$$

where $k_B$ is the Boltzmann constant and *n* is the number of atoms per formula unit. The first term corresponds to the phonon energy (atomic contribution), while the second term represents the electronic contribution.

The thermal pressure is linked with the internal thermal energy by Grüneisen parameter $\gamma$ as in the following Equation,

$$P_{th}(V,T) = \frac{1}{V}\int \gamma(V,T)dE_{th}(V,T). \tag{7}$$

We employ the highest temperature of 8000 K in the present calculations as a reference temperature $T_0$ in order to constrain the reference isotherm as tightly as possible within the broad pressure range. Although in the previous study [7], the following functional form for $\gamma(V)$

$$\gamma(V) = \gamma_0\left\{1 + a\left[(V/V_0)^b - 1\right]\right\} \tag{8}$$

was employed, we realized that three additional parameters $\gamma_0$, *a*, and *b* make the least-square analyses less stable and less systematic. Instead, in this study, we assume the $\gamma$ to be constant for each composition. The previous study [7] reported that the variation of $\gamma$ of pure Fe is 0.2 only from 100 to 400 GPa and from 4000 and 7000 K and we confirmed that this small variation of $\gamma$ does not affect the results of analyzed thermoelasticity. Consequently, the present EoS model requires six parameters in total ($V_0$, $K_{T_0}$, $K'_{T_0}$, $\gamma$, $e_0$, and $g$) to calculate *P* at a given *V*, *T*. These parameters are determined by least squares analyses on the datasets obtained from the AIMD calculations. The derived EoS parameters for best-fit compositions are summarized in Table S1.

Derivative quantities of EoS such as $K_T$ and $\alpha$ are obtained based on the thermodynamic definitions as $(\partial P/\partial V)_T = -K_T/V$ and $(\partial P/\partial T)_V = \alpha K_T$, respectively. $K_T$ is then converted to $K_S$ as

$$\frac{K_S}{K_T} = 1 + \alpha\gamma T, \tag{9}$$

and adiabatic temperature gradient is computed using the relationship,

$$\left(\frac{\partial T}{\partial P}\right)_S = \frac{\alpha VT}{C_P} = \frac{\gamma T}{K_S}. \tag{10}$$

$V_P$ is then calculated for each composition (Figure S1) as $V_P = \sqrt{\frac{K_S}{\rho}}$ along two different adiabats explained below.

*4.3. Adiabats*

The adiabatic temperature profile is calculated numerically by integrating Equation (7) from the ICB pressure. $\rho$ and $V_P$ are calculated along the adiabats anchored by two possible ICB temperatures: $T_{ICB}$ = 5000 K and $T_{ICB}$ = 6500 K. The former $T_{ICB}$ is found to give ~3700 K at 136 GPa, which is close to a proposed core-mantle boundary temperature [38,39]. The latter corresponds to the melting temperature ($T_M$) of pure iron at 329 GPa [17,18], which would be close to the upper bound of ICB temperature since $T_M$ of iron-LE alloys are expected in general to be lower than the $T_M$ of pure Fe.

**Supplementary Materials:** The following are available online at http://www.mdpi.com/2075-163X/10/1/59/s1, Figure S1: The calculated P-V-T data of iron alloys with fitted EoS. Filled red circles represent the data used for EoS analysis, Table S1: EoS parameters for the best-fit composition models.

**Author Contributions:** H.I. and T.T. conducted the ab initio calculations. Both authors analyzed the results and wrote the manuscript. All authors have read and agreed to the published version of the manuscript.

**Funding:** This research was supported by X-ray Free Electron Laser Priority Strategy Program (MEXT), and KAKENHI JP15H05834, JP26800237, JP17K05638, JP17H06457, and JP26287105.

**Acknowledgments:** We thank M. Ohsumi for helping with the analyses, C. Shiraishi for helping with the data plotting, and S. Kaneshima and in particular T. Ohtaki for the helpful discussion.

**Conflicts of Interest:** The authors declare no conflicts of interest.

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
