# Peer review of "Ab Initio Thermoelasticity of Liquid Iron-Nickel-Light Element Alloys"

_minerals, doi:10.3390/min10010059_

Round 1

Reviewer 1 Report

The authors performed massive computations of liquid Fe alloys with various compositions under whole outer core P,T conditions to obtain the thermoelastic properties. Their results generally agree with previous reports but are more comprehensive. I would like to recommend it for publication in Minerals after major revision.

The discussion on bulk earth composition is particularly interesting, but it will be more readable and clear if the treatment of calculations can be more specific in the text, e.g., how the Mg/Si Mg/Fe ratios are obtained.

I suggest the authors address more on the ICB density jump since this is an important constraint and most data of solids are available from literatures.  

The authors used ultrasoft potential, distinct from normally adopted more accurate PAW potential. How is that compared to experimental eos? Besides, the authors also discriminated the empirical pressure correction adopted in Badro 2014, but there is no explicit clarification of the problem: is this correction unnecessary? and the method used in this study is already matching the experiments or can prove experimental wrong?

There are no uncertainties reported in Figures and Tables, but I think it is VERY necessary to report that and include them in discussions.

It is found the misfits of Fe(-Ni)-C system is distinctly large and the authors concluded carbon can be eliminated from major LE in the OC. This may not be true if you include the ternary or higher order multicomponent system, so I think this is an inappropriate conclusion.

Please specify which code was used, k points, etc.

Author Response

Comment 1: The authors performed massive computations of liquid Fe alloys with various compositions under whole outer core P,T conditions to obtain the thermoelastic properties. Their results generally agree with previous reports but are more comprehensive. I would like to recommend it for publication in Minerals after major revision.

Reply: We appreciate Reviewer #1’s positive comment to our work. We revised the manuscript appropriately considering all the comments.

Comment 2: The discussion on bulk earth composition is particularly interesting, but it will be more readable and clear if the treatment of calculations can be more specific in the text, e.g., how the Mg/Si Mg/Fe ratios are obtained.

Reply: The Mg/Si and Mg/Fe ratios were estimated using the total mass of the Earth, weight and atomic % of major elements (Mg, Fe, Si, and O) of the pyrolytic model for the mantle, and the optimized composition models for the core. We mention this in the revised text (l. 191-193).

Comment 3: I suggest the authors address more on the ICB density jump since this is an important constraint and most data of solids are available from literatures.

Reply: It is true that the ICB density jump is a key quantity. Large density contrasts between the outer and inner core (more than 5%) at the ICB reported in seismological studies (Dziwonski and Anderson, 1981) suggest major light elements in the core must be partitioned more into liquid iron than into solid iron. Although O would be a strong candidate which shows this behavior, it is hard to place any conclusion since the melting phase relations in the Fe-light element systems at the ICB pressure (328 GPa) remain unclear. So as already mentioned in the original manuscript, it is substantial to clarify the melting phase relations of Fe-light element systems at the ICB pressure.

Comment 4: The authors used ultrasoft potential, distinct from normally adopted more accurate PAW potential. How is that compared to experimental eos? Besides, the authors also discriminated the empirical pressure correction adopted in Badro 2014, but there is no explicit clarification of the problem: is this correction unnecessary? and the method used in this study is already matching the experiments or can prove experimental wrong?

Reply: Ultrasoft pseudos have the same level of transferability as PAW potentials if the electronic configurations considered are not quite peculiar, as in this study, and the settings are chosen appropriately. Our ultrasoft pseudos have been examined extensively in our previous calculations we already published (Tsuchiya et al., Phys. Rev. Lett., 2016; Tsuchiya and Fujibuchi, Phys. Earth Planet. Int., 2009; Ichikawa et al., J. Geophys. Res., 2014; Ichikawa and Tsuchiya, Phys. Earth Planet. Int., 2015). In particular, properties calculated using our puseudopotentials have been compared with those obtained using the all-electron technique in detail in Tsuchiya and Fujibuchi (2009) and both are in excellent agreement with each other. Conversely, even PAW potentials could give worse results when one choses inappropriate settings. So we believe that the quality of our calculations is not worse than results obtained using PAW. We do not compare calculated EoS of liquids with experiments because it is unclear how experimental EoS of liquids are reliable. But we also compared calculated EoS of solid iron with experiments and earlier calculations in detail and found that our EoS agree excellently with the others (see Tsuchiya and Fujibuchi, 2009). As we mention in the original text, the misfits and optimized compositions can be changed depending on the pressure corrections. But it is unclear how we can determine the corrections accurately without any empirical procedure particularly at multi-megabar pressures. To exclude any biases, we decided not to apply any empirical corrections.

Comment 5: There are no uncertainties reported in Figures and Tables, but I think it is VERY necessary to report that and include them in discussions.

Reply: We add errors which primarily come from the fitting procedures in Tables. Uncertainties in densities and velocities are already plotted in original Fig. 2 by shades. On the other hand, errors in misfits coming from the fitting procedures are comparable (or even smaller) to the size of symbols in Fig. 1. We mention this in the revised caption of Fig. 1.

Comment 6: It is found the misfits of Fe(-Ni)-C system is distinctly large and the authors concluded carbon can be eliminated from major LE in the OC. This may not be true if you include the ternary or higher order multicomponent system, so I think this is an inappropriate conclusion.

Reply: So far, no study showing the elasticity of liquid Fe-C alloy drastically changes by adding a second light element at the outer core pressures has been reported. Therefore, it is unclear if the situation changes for ternary or higher-order multicomponent systems or not. Maybe this is less likely. But we have not yet confirmed this, so we add in l. 132-133 a sentence “, though there is possibility that the situation could change in ternary or higher-order multicomponent systems.”

Comment 7: Please specify which code was used, k points, etc.

Reply: AIMD was performed using a PWSCF code [Giannozzi et al., 2009] for electronic structure with an original implementation of the constant temperature molecular dynamics (MD) module [Usui and Tsuchiya, 2010] as implicitly explained in the original method section. (All the information can be obtained from the references cited.) Liquids in principle have no periodic structure, thus the Γ point only is sampled in our simulations. This is a general treatment for simulations of liquids. We add these in the revised text in l. 226-228 and l. 254-255.

Reviewer 2 Report

The authors studied the thermoelasticity of the liquid alloys in outer core conditions. This work is significant to the Earth’s core science. I can recommend it to be published on Minerals. However, as a researcher in ab initio field, I have some suggestions.

I hope to see more details of method, e.g. which code package they implemented, what is the time step, how long of the AIMD simulation, how many atoms in every unit cell,  how large is the fluctuation of temperature and pressure profile versus the simulating time, etc. What is the x-axis represent in Figure 1, percentage? The color in Figure 2. is not clear enough to easily identify.

Author Response

Comment 1: I hope to see more details of method, e.g. which code package they implemented, what is the time step, how long of the AIMD simulation, how many atoms in every unit cell,  how large is the fluctuation of temperature and pressure profile versus the simulating time, etc.

Reply: Reply: AIMD was performed using a PWSCF code [Giannozzi et al., 2009] for electronic structure with an original implementation of the constant temperature molecular dynamics (MD) module [Usui and Tsuchiya, 2010] as implicitly explained in the original method section. (All the information can be obtained from the references cited.) As partially mentioned already in the original method section, MD cells basically contain 50 atoms as in Ichikawa et al. (2014) but 100 atoms for the optimized compositions. Standard deviations in calculated T and P are found to be ~50 K and ~3 GPa at 5000 K and ~130 GPa and ~120 K and ~6 GPa at 8000 K and ~400 GPa, respectively. We add these in the revised text in l. 226-228, l. 236-237, and l. 257-259.

Comment 2: What is the x-axis represent in Figure 1, percentage? The color in Figure 2. is not clear enough to easily identify.

Reply: The x-axis in Figure 1 represents atomic fractions of light elements as described in the caption. The velocities and densities of the optimized compositions are mostly overlapped, so it is not easy to distinguish them even though any color is assigned. But the important point is the fact that they have only the small differences. Figure 2 does show this.

Reviewer 3 Report

Please, see attached report

Author Response

Comment 1: The paper in question aims to provide a contribution to get insight into the role of light elements (C/H/Si/O) as supposed chemical components of the outer core. The Authors use Ab-initio molecular dynamics to predict the trends of some relevant observables (VP and ρ) of liquid Fe-Ni-X alloys as a function of X (X=C/O/Si/H). They then determine the Fe-Ni-X composition that best fits the PREM model. This way, the Authors infer information about the tendency of a given light element among those investigated to enter the Fe-Ni based alloy. Moreover, the Authors draw conclusions about the bulk earth’s composition.

I think the manuscript under reviewing positively contributes to help understand a complex and largely unknown phenomenology, occurring throughout the outer core and reflecting upon any model of the bulk earth composition. The approach the Authors have followed is correct and appropriate to tackle the problem under investigation.

The paper is well organized and I recommend it for publication with Minerals, provided that the points listed below are properly fixed:

Reply: We really appreciate Reviewer #3’s positive comments to our work. We revised the manuscript appropriately considering all the comments.

Comment 2: 1) the argument developed by the Authors pivots around PREM, which is implicitly assumed to be the target they aim to reproduce. I am wondering how the use of the AK135 model (Kennett, Engdah and Buland, Geophys. J. Int., 122:108-124, 1995; Montagner and Kennett, Geophys. J. Int., 125: 229-248, 1996) would change the conclusions. I think it should not affect them significantly, but this point is to be considered. By the way, the AK135 model is supposed to be particularly appropriate to describe the core. Let me rephrase this paragraph as “what about the sensitivity of the present model to its target, given that the target is not necessarily unique?”

Reply: As Reviewer 3 states, the velocity structure of the Earth’s interior depends on the reference model. It can be identified that the AK135 model has the P wave velocity different from the PREM model in particular at the uppermost and lowermost outer core. Contrasts between the AK135 and PREM reach ~0.11 km/s and ~0.065 km/s at the uppermost and lowermost part, respectively. As Reviewer 3 guesses, these differences do not make any significant changes in the insights obtained from our analyses. We add this discussion in the revised text (l. 176-182).

Comment 3: 2) Correlations between light elements have been neglected. I do realize this choice reflects a compromise between computing time and complexity of the parameter space. The Authors should yet mention such aspect and comment whether/how correlations between light elements simultaneously entering the liquid Fe-Ni-X-alloy affect their results. In this view, I believe the Authors have shed light upon general tendencies of light elements and their related content limits, rather than determined absolute compositions.

Reply: We investigate in this study Fe-light element alloy liquids with a single light element because computations of higher-order multicomponent systems cost considerably and also we believe that the elasticity of alloy liquids is controlled primarily by the major light element. But this is just an assumption. So it is important to clarify the properties of Fe-light element alloy liquids with multiple light elements in future works. We add this in the revised text (l. 219-221).

Comment 4: All the more, the equations from 122 to 126 row provide trends that correlate Fe-Ni-X’s compositions with inner-outer core temperature, and should not be taken as physical parametrizations of actual elemental contents. By the way, as far as I could understand the Authors used two T-points to fit linear functions (122-126 rows). This voids the obtained expressions of any physically pithy bearing. I suggest the Authors could tune their discussion accordingly, paying an extra care to the geochemical meaning they ascribe to Xelements, in general.

Reply: The density of Fe-light element alloys is sensitive to temperature but the velocity is not. So the amounts of light elements required to reproduce the seismological values increase with decreasing temperature. The equations Reviewer 3 points simply indicate this relation for each light element specifically as mentioned in the text.

Comment 5: 3) I think table 2 is not very useful to help a reader understand the text. The misfit values are set out in Figure 1, already, and their absolute values are comparatively little effective to corroborate the arguments discussed. I suggest the Authors to think about whether removing Table 2 at all, and replacing it with a Figure (or another Table, as well), which helps follow the discussion on how the bulk earth’s composition changes because of the selective capacity of the outer core to incorporate light elements.

Reply: Table 2 describes the specific formula of the best-fit compositions with their misfits. It is not easy to pick up these from Fig. 1. So we decide to leave Table 2 in the main text.

Comment 6: 4) The elemental outer core’s composition is related to the light elements partitioning over the major lower mantle mineral phases. A point that should be considered, though it is still little investigated. In this light, with reference to H, for instance, I would mention Merli et al (2017) and Hernàndez et al (2013). Hernàndez, Alfè, Brodholt (2013) Earth Planet. Sci. Lett. 364, 37–43 Merli, Bonadiman, Diella, Pavese (2017) Geochimica et Cosmochimica Acta 173 (2016) 304–318

Reply: The outer core composition should be related to the light element partitioning between silicate melt (magma ocean) and liquid iron expected to occur in the early Earth (e.g., Wood et al., 2006, Nature) rather than the partitioning over the major lower mantle phases. Incorporation of light elements associated with the reactions of the lower mantle minerals and the outer core iron liquid may rather form a light element rich layer, which is stably stratified due to a small density at the upper part of the core (less than 100 km thickness even after 4.5 billion years since the core formation) and unlikely to affect the chemistry of the whole outer core (see Buffett and Seagle, 2010, J. Geophys. Res.).

Round 2

Reviewer 1 Report

The authors have addressed my concerns. I recommend it to publish. One minor comment is that I do not think it is appropriate to say "Calculated results indicate that Si and S counterintuitively have larger effects on the density and P-wave velocity of liquid iron than O, C". C and O have smaller volumes and are expected for such a behaviour (Posner 2019).

Author Response

Comment: The authors have addressed my concerns. I recommend it to publish. One minor comment is that I do not think it is appropriate to say "Calculated results indicate that Si and S counterintuitively have larger effects on the density and P-wave velocity of liquid iron than O, C". C and O have smaller volumes and are expected for such a behaviour (Posner 2019).

Reply: Thank you very much for your careful reading on our revised manuscript. Your comment is quite helpful to further improve the manuscript. The calculated results (Table 1) clearly show that to reproduce the seismological density of the outer core, smaller amounts of Si and S are required than of O and H, even though masses of O and H are much smaller than of Si and S. This behavior seems somewhat counterintuitive but quite reasonable if considering the smaller sizes of O and H as the reviewer points out and as mentioned in the original text (Sect 2.1). On the other hand, this is not so distinct for C. This would be because volume and mass contractions by the C incorporation is mostly cancelled out. Considering these, we change the sentence in the abstract and l. 71-72 to "the effects of Si and S incorporations on ρ are larger than those of O and H".  

Since having missed finding the recent study by Posner 2019 suggested by the reviewer, we cite it in the revised manuscript and mention "These behaviors are consistent with a recent study reporting structural and dynamical properties of Fe-LE alloy liquids." in l. 72-74.